# Implantable Cardioverter Defibrillator Multisensor Monitoring during Home Confinement Caused by the COVID-19 Pandemic

**DOI:** 10.3390/biology11010120

**Published:** 2022-01-12

**Authors:** Matteo Ziacchi, Leonardo Calò, Antonio D’Onofrio, Michele Manzo, Antonio Dello Russo, Luca Santini, Giovanna Giubilato, Cosimo Carriere, Vincenzo Ezio Santobuono, Gianluca Savarese, Carmelo La Greca, Giuseppe Arena, Antonello Talarico, Ennio Pisanò, Massimo Giammaria, Antonio Pangallo, Monica Campari, Sergio Valsecchi, Igor Diemberger

**Affiliations:** 1Institute of Cardiology, Department of Experimental, Diagnostic and Specialty Medicine, University of Bologna, Policlinico S.Orsola-Malpighi, 40138 Bologna, Italy; igor.diemberger@unibo.it; 2Policlinico Casilino, 00118 Rome, Italy; leonardocalo.doc@gmail.com; 3“Unità Operativa di Elettrofisiologia, Studio e Terapia delle Aritmie”, Monaldi Hospital, 80131 Naples, Italy; donofrioant1@gmail.com; 4OO.RR. San Giovanni di Dio Ruggi d’Aragona, 84125 Salerno, Italy; mikemanzo2002@libero.it; 5Clinica di Cardiologia e Aritmologia, Università Politecnica delle Marche, “Ospedali Riuniti”, 60126 Ancona, Italy; antonio.dellorusso@gmail.com; 6“Giovan Battista Grassi” Hospital, 00122 Rome, Italy; lsantini@alice.it; 7“F. Spaziani” Hospital, 03100 Frosinone, Italy; giovannagiubilato@gmail.com; 8Azienda Ospedaliera Universitaria Ospedali Riuniti di Trieste—Cattinara, 34149 Trieste, Italy; cosimo.carriere@libero.it; 9Policlinico di Bari, University of Bari, 70126 Bari, Italy; eziosantobuono@gmail.com; 10S. Giovanni Battista Hospital, 06034 Foligno, Italy; savarese75@yahoo.it; 11Fondazione Poliambulanza, 25124 Brescia, Italy; carmelo.lagreca@poliambulanza.it; 12Ospedale Civile Apuane, 54100 Massa, Italy; giuseppe.arena@uslnordovest.toscana.it; 13SS. Annunziata Hospital, 87100 Cosenza, Italy; antonello.talarico@libero.it; 14Vito Fazzi Hospital, 73100 Lecce, Italy; enniopisano@hotmail.com; 15Division of Cardiology, Maria Vittoria Hospital, 10144 Turin, Italy; massimo.giammaria@aslcittaditorino.it; 16Grande Ospedale Metropolitano “Bianchi-Melacrino”, 89124 Reggio Calabria, Italy; panganto@gmail.com; 17Boston Scientific Italia, 20134 Milan, Italy; monica.campari@bsci.com (M.C.); sergio.valsecchi@bsci.com (S.V.)

**Keywords:** ICD, CRT, heart failure, remote monitoring, multisensor

## Abstract

**Simple Summary:**

During the COVID-19 pandemic, utilization of remote monitoring platforms was recommended. The HeartLogic algorithm identifies patients at risk of heart failure events, combining multiple sensors available on implantable cardioverter defibrillators. This analysis examined how multiple CIED sensors behave in periods of anticipated restrictions pertaining to physical activity. We demonstrated a significant drop in median activity level immediately after the implementation of stay-at-home orders, whereas there was no difference in the other contributing sensors. The weekly rate of heart failure alerts was significantly higher during the lockdown and post-lockdown than that reported in the pre-lockdown.

**Abstract:**

Aims: The utilization of remote monitoring platforms was recommended amidst the COVID-19 pandemic. The HeartLogic index combines multiple implantable cardioverter defibrillator (ICD) sensors and has proved to be a predictor of impending heart failure (HF) decompensation. We examined how multiple ICD sensors behave in the periods of anticipated restrictions pertaining to physical activity. Methods: The HeartLogic feature was active in 349 ICD and cardiac resynchronization therapy ICD patients at 20 Italian centers. The period from 1 January to 19 July 2020, was divided into three phases: pre-lockdown (weeks 1–11), lockdown (weeks 12–20), post-lockdown (weeks 21–29). Results: Immediately after the implementation of stay-at-home orders (week 12), we observed a significant drop in median activity level whereas there was no difference in the other contributing parameters. The median composite HeartLogic index increased at the end of the Lockdown. The weekly rate of alerts was significantly higher during the lockdown (1.56 alerts/week/100 pts, 95%CI: 1.15–2.06; IRR = 1.71, *p* = 0.014) and post-lockdown (1.37 alerts/week/100 pts, 95%CI: 0.99–1.84; IRR = 1.50, *p* = 0.072) than that reported in pre-lockdown (0.91 alerts/week/100 pts, 95%CI: 0.64–1.27). However, the median duration of alert state and the maximum index value did not change among phases, as well as the proportion of alerts followed by clinical actions at the centers and the proportion of alerts fully managed remotely. Conclusions: During the lockdown, the system detected a significant drop in the median activity level and generated a higher rate of alerts suggestive of worsening of the HF status.

## 1. Introduction

The spread of the coronavirus disease 2019 (COVID-19) epidemic required a rapid response. The primary modes of disease prevention have involved limiting exposure and social distancing. At various times, stay-at-home orders were issued in many geographies and were shown to result in a rapid global reduction in physical activity [1]. Physical activity is an important determinant of health [2,3]. During the COVID-19 pandemic, hospital admissions for acute cardiac conditions markedly declined [4,5,6] as access to care was impacted by limited hospital resources and by the reluctance of patients to go to hospital. Guidance from the European Society of Cardiology (ESC) for the diagnosis and management of cardiovascular disease during the COVID-19 pandemic [7] encouraged centers to consider telemedicine to provide patients medical advice and follow-ups. For patients with cardiovascular implantable electronic devices (CIEDs), in-person office visits had to be replaced by remote contact, using the device information obtained through remote monitoring [8,9]. 

To date, scant data exist on the consequences of the home confinement on the clinical status of heart failure (HF) patients with CIEDs [10]. Moreover, it is not known whether the physical activity decrease was associated with a worsening of physiological parameters. Modern CIEDs continuously monitor multiple clinical variables, in order to provide an early warning of changes in the clinical status. Therefore, we examined how multiple CIED sensors behave in the periods of anticipated restrictions pertaining to physical activity.

## 2. Materials and Methods

The home confinement for the COVID-19 pandemic in Italy was imposed from March 8th to May 18th. For this analysis we identified all of the HF patients with reduced left ventricular ejection fraction (≤35% at the time of implantation) who had received a HeartLogic-enabled implantable cardioverter defibrillator (ICD) or cardiac resynchronization therapy ICD (CRT-D) (RESONATE family, Boston Scientific) in accordance with standard indications [11] at 20 study centers (full list of participant centers in Appendix A). Patients had to be on regular monitoring in the LATITUDE (Boston Scientific) platform, with diagnostic data available from at least 1 January 2020. As initialization is required, the HeartLogic index does not become available until 30–37 days after implantation. Thus, in this analysis, we included only devices implanted before November 2019. Before the lockdown, patients were followed up in accordance with the standard practice of the participating centers, based on current international recommendations [12]. The study protocol did not mandate any specific intervention algorithm and physicians were free to remotely implement clinical actions, to schedule extra in-office visits, or to adopt an active monitoring approach. Data were collected at the study centers in the framework of a prospective registry (Rhythm Detect Registry, ClinicalTrials.gov Identifier: NCT02275637) approved by the Institutional Review Board of the Coordinating Center (IRCCS Policlinico S. Matteo, Pavia, Italy) and all participant centers. The research was performed in accordance with the Declaration of Helsinki. All patients provided written informed consent for data storage and analysis. 

### 2.1. HeartLogic Index 

The details of the HeartLogic algorithm have been reported previously [13]. Briefly, the algorithm combines data from multiple sensors: accelerometer-based first and third heart sounds, intrathoracic impedance, respiration rate, the ratio of respiration rate to tidal volume, night heart rate, and patient activity. Each day, the device calculates the degree of worsening in sensors from their moving baseline and computes a composite index. An alert is issued when the index crosses a programmable threshold. 

### 2.2. Design of Analysis

We assessed the trend of all HeartLogic sensors from 1 January to 19 July. The period was divided in 3 phases: pre-lockdown (weeks 1–11), lockdown (weeks 12–20), post-lockdown (weeks 21–29). We calculated the change in all variables among phases, as well as the rate of alerts occurred, and the clinical actions or extra in-office visits performed to manage them. We also compared the sensed parameters that contributed to the calculation of the HeartLogic index in case of alerts among study phases.

### 2.3. Statistical Analysis

Descriptive statistics are reported as means ± SD for normally distributed continuous variables, or medians with 25th to 75th percentiles in the case of skewed distribution. Normality of distribution was tested by means of the nonparametric Kolmogorov–Smirnov test. Categorical data were expressed as percentages. Differences between continuous variables were performed using a Student’s T test for Gaussian variables, and a Mann–Whitney U test non-parametric test for non-Gaussian variables. Differences in proportions were compared by applying chi-squared analysis or Fisher’s exact test, as appropriate. Incidence rates with confidence intervals were calculated and compared, together with the incidence rate ratio. One-way analysis of variance for repeated measures was used to test for differences among phases. A *p* value < 0.05 was considered significant in all tests. All statistical analyses were performed by means of R: a language and environment for statistical computing (R Foundation for Statistical Computing, Vienna, Austria).

## 3. Results

In the analysis, we included 349 ICD and cardiac resynchronization therapy ICD patients who had received the device at the study centers before November 2019. Table 1 shows the baseline clinical variables of all patients in the analysis. No patients developed COVID-19 during the study period.

### 3.1. HeartLogic and Contributing Sensors Trends

Figure 1 shows the weekly averages of the HeartLogic combined index, and all of the physiologic parameters collected by the devices from 1 January to 19 July. Immediately after the implementation of stay-at-home orders (from week 12), we observed a significant drop in the activity level that persisted until week 19, whereas there was no difference in the other contributing sensors. The composite HeartLogic index significantly increased at the end of the lockdown phase (from week 20), and the increase in the average index remained significant until week 28 in the post-lockdown phase.

### 3.2. HeartLogic Alerts, Characteristics and Management

The HeartLogic index crossed the threshold value 35 times in the pre-lockdown phase, 49 times during the lockdown phase (incidence rate ratio 1.71 [95% CI: 1.09–2.72] versus the pre-lockdown; *p* = 0.014), and 43 times during the post-lockdown phase (incidence rate ratio 1.50 [95% CI: 0.94–2.42] versus the pre-lockdown; *p* = 0.072) (Table 2).

Of the 127 reported HeartLogic alerts, 113 (89%) did not require extra in-office visits and were managed remotely. Alert-triggered actions (e.g., drug adjustments, educational interventions) were reported in 34 (27%) cases. In the remaining cases, physicians adopted an active monitoring approach, intensifying the frequency of contacts but not intervening proactively. The proportion of alerts managed remotely, as well as the proportion of alerts triggering clinical actions, remained constant among study phases (Table 2). The average duration of the in-alert state was similar in the three study phases, as well as the maximum value of the index (Table 2). This was confirmed by comparable trends of HeartLogic index during the weeks immediately before and after the alert onset among study phases (Figure 2). On comparing the trends of all physiologic parameters among study phases, we noticed similar values, i.e., comparable contribution from all sensors to the combined HeartLogic index, except for higher activity levels before and after the alert onset in the post-lockdown phase than in the other phases.

## 4. Discussion

In this analysis, we showed a marked decrease in the ICD-measured physical activity, whereas no changes were detected in the other sensors, i.e., heart sounds, intrathoracic impedance, respiration parameters, heart rate. Indeed, the home confinement had no significant impact on single physiologic parameters beyond activity, but we observed an increased number of device-defined HF events, as detected by the combined HeartLogic index. This occurred at the end of the lockdown period and seemed to persist for some weeks after the end of home confinement.

The decline in physical activity during the lockdown was significant, as demonstrated through the use of smartphone accelerometers and algorithms for step counting in the general population [1] and in a small HF group [14]. In the present analysis, we confirmed this finding in a large HF population, although a low impact was expected in these patients, given the lower baseline activity level associated with their reduced functional capacity. The clinical relevance of measured physiologic parameters has been proven [13], as well as the meaningful association between individual sensors with changes in cardiac systolic and diastolic function [15], functional status [16], congestion [17], and prognosis [18,19]. Mitter et al. reported small changes in some parameters at the time of the lockdown in a smaller population of ICD and CRT-D patients [20]. Specifically, they demonstrated a decline in heart rate, an increase in intrathoracic impedance and a decrease in S3, and they interpreted this as a possible improvement linked to a decreased autonomic tone with less activity and potentially less frequent access to unhealthy food options. In our analysis, the night heart rate did not change; indeed, this parameter is a surrogate of resting heart rate and thus not directly associated with the activity level. Moreover, we did not confirm the increase in impedance described by Mitter et al. [20]. It is known that increases in CIED-measured impedance may not only be suggestive of less pulmonary congestion in HF, but may also be associated with the healing of the pocket hematoma or with the left ventricle volume changes induced by CRT early after implantation [21,22]. At the time of the lockdown, all our patients had received the device for at least 5 months; therefore, any effect linked with the initiation of the therapy had plausibly ended. The same applies to first and third heart sound amplitudes. They may improve early after CRT initiation, and we did not notice significant changes at the time of the home confinement in a population of patients implanted months before.

The rate of HeartLogic alerts was higher during the lockdown, as well as the average index after 8 weeks of home confinement. The analysis of the individual sensors during the weeks before the alert onset showed comparable contribution from all sensors to the combined HeartLogic index among study phases, except for higher activity levels in the post-lockdown phase. This suggests that the trigger mechanism of HF decompensation is the same under different activity conditions. Moreover, the decline in activity alone cannot lead to an increase in the index [13], but lower levels of activity could make the system more sensitive to other parameters. However, this would have resulted in different values among phases. Therefore, we tend to believe that the clinical relevance of diagnosed events was comparable among phases, and that patients may have experienced more frequent episodes of true HF decompensation. Indeed, the detrimental effects of the lockdown in patients with cardiovascular diseases have recently been demonstrated [23]. Our findings seem to disagree with the observations of a recent work that suggested potential beneficial effects of the lockdown in ICD patients for the reduction in real-life stressors [24].

In our centers with experts in the remote management of alerts, the COVID-19 restrictions had no impact on their standard practice. Indeed, the proportion of alerts managed remotely and alerts followed by clinical actions did not change between different phases of the pandemic. Consequently, the severity of the individual events was equivalent, since the in-alert states had similar durations and extents.

In previous works, the ability of HeartLogic to detect actionable HF events has been demonstrated, facilitating effective remote management [25,26,27,28,29]. In the context of the COVID-19 pandemic, the multisensor diagnostic platform was used effectively to facilitate the remote assessment of patient conditions. These findings confirm and extend previous anecdotal cases in which the HeartLogic algorithm provided critical data that allowed for the appropriate triage of patients, with reductions in unnecessary clinic visits during the COVID-19 pandemic [30]. 

The main limitation of this study is its observational non-randomized design. Moreover, as mentioned above, no predetermined actions were prescribed in response to HeartLogic alerts or to the individual subject’s reported signs or symptoms.

## 5. Conclusions

The HeartLogic multisensor platform detected the decrease in activity, although the home confinement had no impact on the other sensors. The increased number of alerts during the lockdown suggests that the home confinement had a negative effect on patients’ outcome. This must be taken into account in periods of prolonged social distancing and confirms the importance of ensuring safe and secure access to healthcare facilities for everybody who needs continuity of care.

## Figures and Tables

**Figure 1 biology-11-00120-f001:**
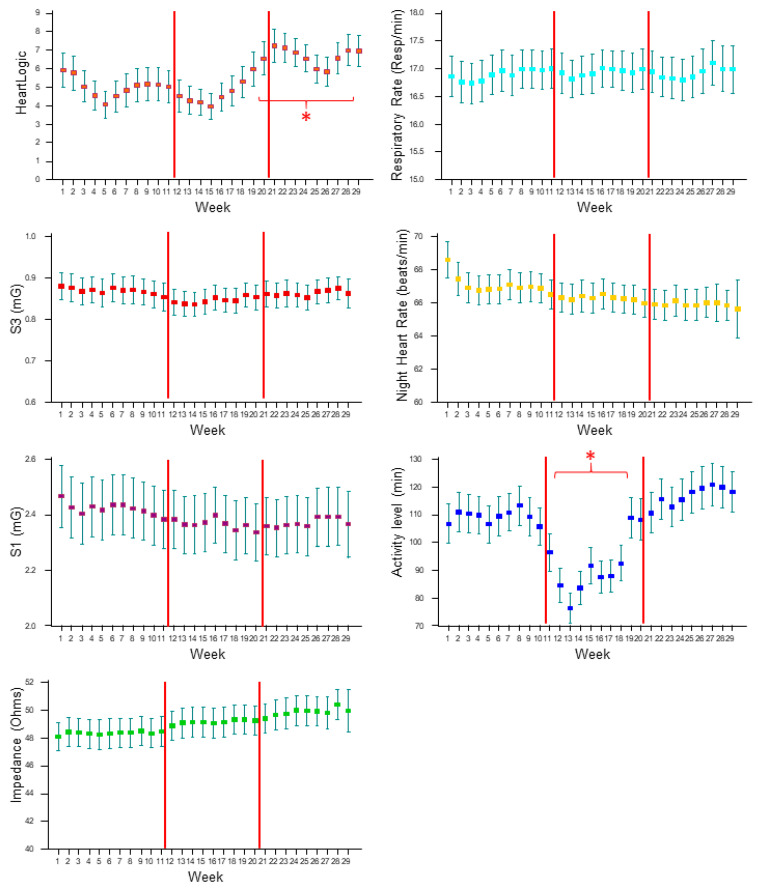
Weekly averages (with standard deviation) of the HeartLogic combined index and all physiologic parameters collected by the devices from 1 January to 19 July. The period was divided in 3 phases: pre-lockdown (weeks 1–11), lockdown (weeks 12–20), post-lockdown (weeks 21–29). S1: First heart sound; S3: Third heart sound; *: *p* < 0.05 versus pre-lockdown average.

**Figure 2 biology-11-00120-f002:**
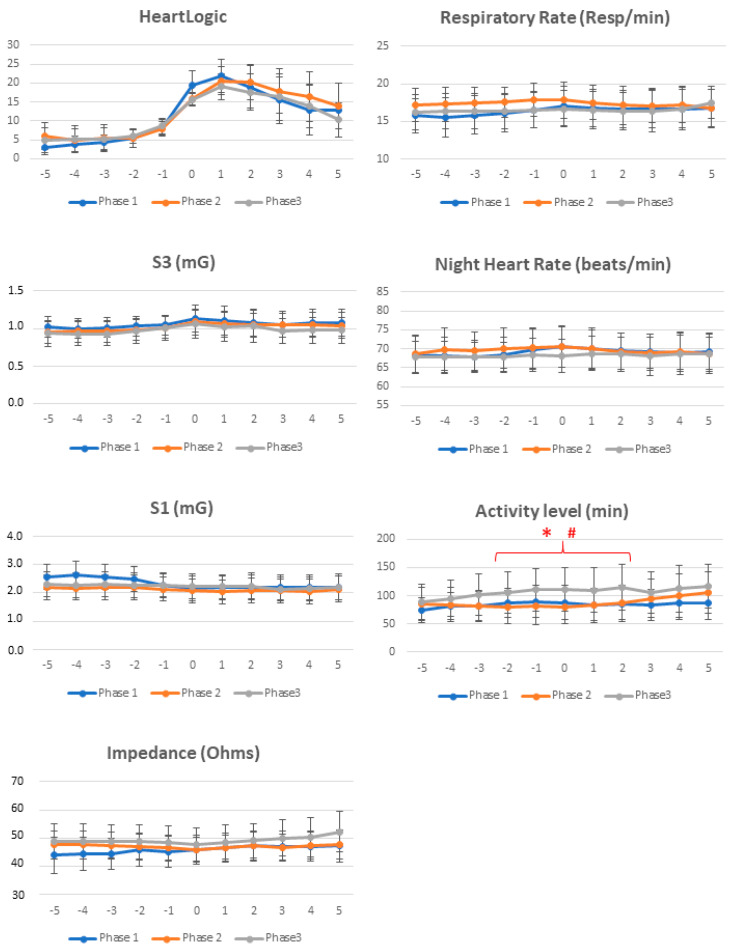
Comparison of trends collected immediately before and after the alerts among study phases. Weekly averages (with standard deviation) of HeartLogic index and all physiologic parameters are reported from 5 weeks before to 5 weeks after the alert onset (week 0). S1: First heart sound; S3: Third heart sound; *: *p* < 0.05 post-lockdown versus pre-lockdown; #: *p* < 0.05 post-lockdown versus lockdown.

**Table 1 biology-11-00120-t001:** Demographics and baseline clinical parameters of the study population.

Parameter	Total N = 349
Male gender, *n* (%)	283 (81)
Age, years	69 ± 11
Ischemic etiology, *n* (%)	156 (45)
NYHA class	
− Class I, *n* (%) − Class II, *n* (%) − Class III, *n* (%) − Class IV, *n* (%)	21 (6) 188 (54) 131 (37) 9 (3)
LV ejection fraction, %	31 ± 8
AF history, *n* (%)	133 (38)
Valvular disease, *n* (%)	63 (18)
Coronary artery disease, *n* (%)	166 (48)
Diabetes, *n* (%)	99 (28)
COPD, *n* (%)	59 (17)
Chronic kidney disease, *n* (%)	101 (29)
Hypertension, *n* (%)	210 (60)
β-Blocker use, *n* (%)	329 (94)
ACE-inhibitor, ARB or ARNI use, *n* (%)	321 (92)
MRA use, *n* (%)	209 (60)
Diuretic use, *n* (%)	324 (93)
Antiarrhythmic use, *n* (%)	84 (24)
Anticoagulant therapy use, *n* (%)	142 (41)
Ivabradine use, *n* (%)	28 (8)
CRT device, *n* (%)	269 (77)
Primary prevention, *n* (%)	329 (94)

NYHA = New York Heart Association; LV = Left ventricle; AF = Atrial fibrillation; COPD = Chronic obstructive pulmonary disease; ACE = Angiotensin-converting enzyme; ARB = Angiotensin II receptor blockers; ARNI = Angiotensin receptor–neprilysin inhibitor; MRA = Mineralocorticoid receptor antagonists; CRT = Cardiac resynchronization therapy.

**Table 2 biology-11-00120-t002:** HeartLogic alerts during the study phases.

	Alerts, *n*	Rate [95% CI], Alerts/100 pt-Weeks	Alert Duration, Days	Maximum Index Value	Alerts with Actions	Remote Management
Pre-lockdown (weeks 1–11)	35	0.91 (0.64–1.27)	47 (29–60)	28 ± 11	11 (31%)	31 (89%)
Lockdown (weeks 12–20)	49	1.56 (1.15–2.06)	39 (28–57)	30 ± 15	11 (22%)	44 (90%)
Post-lockdown (weeks 21–29)	43	1.37 (0.99–1.84)	36 (25–57)	24 ± 12	12 (28%)	38 (88%)

## Data Availability

The experimental data used to support the findings of this study are available from the corresponding author upon request.

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
