# Peer review of "Implantable Cardioverter Defibrillator Multisensor Monitoring during Home Confinement Caused by the COVID-19 Pandemic"

_biology, 2022, doi:10.3390/biology11010120_

Round 1

Reviewer 1 Report

The Authors analysed data retrieved from cardiac implantable devices (ICDs/CRT-Ds, equipped with HeartLogic algorithm), monitored remotely (Latitude system), and implanted in 349 heart failure patients. Patients were implanted and followed in 20 italian centres from January 1st to July 19th 2020. The observation period was divided in 3 phases: Pre-Lockdown (weeks 1-11), Lockdown (weeks 12-20), Post-Lockdown (weeks 21-29). Authors found significantly lower level of physical activity in lock-down period (with no change in other variables), and increase in HeartLogic index at the end of lock-down (remaining high in the post-lockdown period). Authors claim, that the system (HeartLogic) was sensitive to the behavioral changes occurred during the lockdown.

First of all, the Authors have to be congratulated for  the enormous work they did analyzing such a huge amount of data. The data collected can be of paramount importance in the context of Covid pandemic in HF patients. However, the idea on how to design this study and how to show and interpret this data seem to be very far from optimal. The authors took aim, which cannot be verified with data they have. And they came to conclusions, that cannot be infered based on analyses they did. Not having data on real “patients behavior” in pre- during- and post-pandemic period, and not having data on remote FU (HF exacerbations), the only thing the study presents (a very interesting, however much weakened by the lack of clinical data and no possibility to assess alert specificity-sensitivity) is how devices sensors and HeartLogic algorithm behaves in the period of anticipated (we do not know for sure if/in what proportion patients obeyed to the rules) restrictions pertaining to physical activity. To be acceptable, the manuscript would have to undergo massive reworking, with clinical data included.

  1. The work needs a significant linguistic revision
  2. Introduction: is much too long, should be more concise and shortening by at least 100 words (360 now) is needed.
  3. Introduction: Aim of the study “we examined whether data collected by multiple CIED sensors before, during and after the period of home confinement for the COVID-19 pandemic may elucidate the possible consequences of physical activity decrease” – you did not examine this (see a point below). The aim has to be modified.
  4. Material and method: the descriptive part (part no 2.) has to be shortened significantly.
  5. Results: inclusion criteria seem unclear. In Results you claim : “On March 8th 2020, the HeartLogic feature was active in 349 ICD and cardiac resynchronization therapy ICD patients, who had received the device at the study centers before November 2019”. In Methods you are saying : “We assessed the trend of all HeartLogic sensors from January 1st to July 19th” and “Patients had to be on regular monitoring in the LATITUDE (Boston Scientific) platform on March 8th 2020, and with diagnostic data available from at least January 1st 2020, i.e. device implanted before November 2019”. This is unclear – can you please explain whom did you include in a more straightforward way?
  6. Results: “Alert-triggered actions (e.g. drug adjustments, educational interventions) were reported in 34 (27%) cases” – how about the remaining cases?
  7. Results: how did it happen that HeartLogic index increased? No other parameters changed. What contributed to this increase?
  8. Results – table 1 – only 57% on ACE-I? With mean EF of 31% and 77% with CRT? Why such a small number? Number of patients on MRA? Any patients on ARNI? Any patients on SGLT2-i? Number of patients on anticoagulation (38% had AF)?
  9. Table 2. The numerically highest number of alert durations was in the pre-lockdown phase. How this can be explained?
  10. Figure 1. While a significantly higher HeartLogic index can be seen only in third time-period (post-lockdown), Authors found a highest number of HeartLogic alerts in the second time-period (P<0.05 versus pre-lockdown), but not in third (P=NS versus pre-lockdown). How can the Authors comment on that?
  11. Figure 1. As can be noted from this figure, activity level decreased indeed during lockdown. However, these constrains in activity did not translate in any way into mean night heart rate (should increase along with deconditioning) and respiratory rate (the same behavior expected). Can you comment on this?
  12. Figure 2. I am not sure what is this figure showing and what is the rationale for showing such data? Why are Authors expecting different behavior of various variables (+-5 weeks to alert) in different time-periods?
  13. Discussion: Is much too long (ca 1000 words), should be shortened by at least ¼. The Authors concentrate on diagnostic sensitivity of the system, which has been not assessed here. Authors claim “Indeed, the home confinement had no significant impact on single physiologic parameters beyond activity, but it still resulted in an increased number of device-defined HF events, as detected by the combined HeartLogic index”. This is contradictory (no change in single parameter, but increase in index). Authors did not assess HF events, they assessed only “device-defined HF events, as detected by the combined HeartLogic index”.
  14. Discussion; “we imagine that some patients might have experienced episodes of HF decompensation as a consequence of the changes in behaviors, and possibly for the discontinuation of chronic treatments” – this is highly speculative. The Authors claim that they followed the patients, while they show no data on HF exacerbations/remote outcomes.
  15. Conclusions: they are not based on what the Authors showed in this manuscript – need significant reworking.

  1. Abstract: you are saying, that you aim was :” examine whether the algorithm may elucidate behavioral changes that impact HF decompensation”. And you assessed indeed, how the HeartLogic reacted during three periods of time (called pre-lockdown, lockdown and post-lockdown). But you did not assess behavioral changes. So, we cannot say, based on what you analyzed, if the system responded appropriately on patients’ behavior (e.g. – maybe the alerts were appropriate, the sensitivity / specificity of the system were OK, but, on the other hand – we can just as easily say it was not. You provide no data to assess that). In the MultiSense study, HeartLogic had a sensitivity of 70% and inappropriate alert-rate of ca 1.47/patient/year (PMID: 28254128). Therefore, this aim has nothing to do with what you actually did.
  2. Abstract: „there was no difference in the other contributing sensors” – I would rather say “no difference in other parameters/variables/characteristics” [measured by specific sensors]. It were not sensors that changed, the parameters were (or were nor) changing.
  3. Abstract: “median composite HeartLogic index increased at the end of Lockdown” – what does the index mean? Must be (albeit very briefly) explained here.
  4. Abstract: Conclusions: “The system was sensitive to the behavioral changes occurred during the lockdown” – how do you know that? How do you know what happened with behavioral changes during lockdown? You did not assess them. We can suspect less mobility etc, but….you did not assess them!
  5. Abstract : “The higher rate of alerts during lockdown suggests the worsening of the HF status” – this is only a speculation, cannot be an inherent part of conclusions. See results of many studies, e.g. MultiSenseYou did not assess the HF status.
  6. Figures 1 and 2. What are S1 and S3? I suspect they represent heart sounds, but it should be explained below figures.

Author Response

Dear Reviewer

enclosed please find the revised version of our manuscript “Implantable Cardioverter Defibrillator Multisensor Monitoring during home confinement caused by the COVID-19 pandemic” which is submitted for publication in Biology.

We are grateful to you and to other  reviewer for the time devoted to review our manuscript.

The manuscript has been revised in the light of the reviewers’ comments and we include with our submission a detailed response to the issues raised.

We are confident that the present revised version of the manuscript will be considered acceptable for publication in Biology.

We are grateful for your kind consideration.

Sincerely yours,

Matteo Ziacchi

Reviewer 2 Report

M. Ziacchi et al examined whether data collected by multiple CIED sensors before, during and after the period of home confinement for the COVID-19 pandemic may elucidate the possible consequences of physical activity decrease, and help identifying behavioral changes that impact HF decompensation. They found that the HeartLogic multisensor platform was sensitive to the behavioral changes occurred during the COVID-19 lockdown and that the weekly rate of alerts was significantly higher during Lock-down and Post-Lockdown than that reported in Pre-Lockdown and that the median duration of alert state and the maximum index value did not change among phases, as well as the proportion of alerts followed by clinical actions at the centers and the proportion of alerts fully managed remotely. The authors conclude that the higher rate of alerts during lockdown suggests the worsening of the heart failure status, possibly explained by the behavioral and treatment changes during the home confinement.

The manuscript is well written. The methods are clear. The conclusion is comprehensive, however, the authors should elaborate more clearly the clinical impact of the findings for our daily life.

What is the learning from this study for possible future lockdowns in this patient population?

Author Response

(The authors gave the same response as above.)

Round 2

Reviewer 1 Report

The Authors have addressed all doubts and questions of this reviewer.

Minor issues:

  1. Minor language editing is still needed (e.g. the implementation of stay at home orders)
  2. The Abstract in the web version differs  from this on the hardcopy
  3. All abbreviations  should be explained before the first use (e.g. ESC, CIED, HF in "Introduction", to name only some)

Author Response

Dear Reviewer,

      Please find enclosed the revised version of our manuscript. We are grateful to you for the time devoted to review our manuscript.

The manuscript has been revised in the light of your comments and we include with our submission a detailed response to the issues raised.

We are confident that the present revised version of the manuscript will be considered acceptable for publication.

Sincerely yours,

Matteo Ziacchi, MD

Answer to Reviewer 1

Minor issues:

1. Minor language editing is still needed (e.g. the implementation of stay at home orders)

The manuscript has been revised as suggested.

2. The Abstract in the web version differs  from this on the hardcopy

The final and correct version of the abstract is the one in the attached file. We have not been able to change the text in the web version at this stage of the revision process.

3. All abbreviations  should be explained before the first use (e.g. ESC, CIED, HF in "Introduction", to name only some)

In the revised manuscript version, all abbreviations have been explained at their first use.
